# Thyroid Cancers: From Surgery to Current and Future Systemic Therapies through Their Molecular Identities

**DOI:** 10.3390/ijms22063117

**Published:** 2021-03-18

**Authors:** Loredana Lorusso, Virginia Cappagli, Laura Valerio, Carlotta Giani, David Viola, Luciana Puleo, Carla Gambale, Elisa Minaldi, Maria Cristina Campopiano, Antonio Matrone, Valeria Bottici, Laura Agate, Eleonora Molinaro, Rossella Elisei

**Affiliations:** Endocrine Unit, Department of Clinical and Experimental Medicine, University of Pisa, 56126 Pisa, Italy; lorussoloredana@hotmail.it (L.L.); virgicap@hotmail.com (V.C.); lau.val@hotmail.it (L.V.); carlottagiani@hotmail.com (C.G.); violadavid@hotmail.it (D.V.); puleo.luciana@gmail.com (L.P.); gambalecarla@libero.it (C.G.); elisaminaldi@hotmail.it (E.M.); cristina.campopiano@gmail.com (M.C.C.); anto.matrone@yahoo.com (A.M.); valeriabottici@gmail.com (V.B.); laura.agate@virgilio.it (L.A.); elemoli@hotmail.com (E.M.)

**Keywords:** differentiated thyroid cancer, medullary thyroid cancer, targeted therapy, tyrosine kinase inhibitors, sorafenib, lenvatinib, vandetanib, cabozantinib, selpercatinib, pralsetinib

## Abstract

Differentiated thyroid cancers (DTC) are commonly and successfully treated with total thyroidectomy plus/minus radioiodine therapy (RAI). Medullary thyroid cancer (MTC) is only treated with surgery but only intrathyroidal tumors are cured. The worst prognosis is for anaplastic (ATC) and poorly differentiated thyroid cancer (PDTC). Whenever a local or metastatic advanced disease is present, other treatments are required, varying from local to systemic therapies. In the last decade, the efficacy of the targeted therapies and, in particular, tyrosine kinase inhibitors (TKIs) has been demonstrated. They can prolong the disease progression-free survival and represent the most important therapeutic option for the treatment of advanced and progressive thyroid cancer. Currently, lenvatinib and sorafenib are the approved drugs for the treatment of RAI-refractory DTC and PDTC while advanced MTC can be treated with either cabozantinib or vandetanib. Dabrafenib plus trametinib is the only approved treatment by FDA for *BRAF^V600E^* mutated ATC. A new generation of TKIs, specifically for single altered oncogenes, is under evaluation in phase 2 and 3 clinical trials. The aim of this review was to provide an overview of the current and future treatments of thyroid cancer with regards to the advanced and progressive cases that require systemic therapies that are becoming more and more targeted on the molecular identity of the tumor.

## 1. Introduction

Thyroid cancer (TC) is the most common endocrine neoplasia and represents approximately 2.9% of all new cancer cases in the United States each year [1]. In the last three decades, the incidence rate of TC has continuously increased all over the world and, although it was mainly attributed to an increased detection rate of small tumors, a real increase of TC was also identified as demonstrated by an increase of larger tumors too [2]. In this review, we discuss the different types of current and future systemic therapies to be used in different types of advanced and metastatic TC. A brief description of the different histotypes of TC and signaling pathways is required to better understand the different types of targeted therapies available.

### 1.1. Differentiated Thyroid Cancer

Differentiated thyroid cancer (DTC) is the most frequent tumor type, representing > 90% of all TC [3]. It originates from follicular cells and includes the three main subtypes: papillary (PTC), follicular (FTC), and Hürthle cell carcinoma (HTC). Despite the overall survival (OS) rate being 98.3% at 5 years for the majority of cases [1], local recurrence (thyroid bed or cervical lymph nodes) occurs in about 20% of patients and distant metastasis in approximately 10%, lungs being the most common site of metastases (50%) followed by bone (25%) [4]. In one third of advanced DTC, the metastatic lesions lose the ability to take up iodine (RAI-refractory DTC) with subsequently no efficacy of radioiodine with ^131^I (RAI) and decrease of OS rate (less than 10% at 10 years) [5]. As well as RAI-refractoriness, the stage of neoplastic disease at diagnosis can also predict mortality. In particular, according to the 8th Edition of the American Joint Committee on Cancer TNM classification [6], an excellent prognosis is reported for stage I and II of DTC with an overall disease-specific survival of >75% to 95% at 10 years that falls to 60% to <50% for stage III and IV, respectively.

Poorly differentiated thyroid cancer (PDTC) is a heterogeneous group that includes those TCs that, according to the Turin classification, lose the papillary nuclear features and have a solid, insular, or trabecular growth pattern with an increased number of mitoses and necrosis [7,8]. PDTC in respect with other DTCs has a higher risk of persistence/recurrence both in the neck and at distant localization (lung, liver, bone, and brain) and a higher mortality [9,10].

According to the most recent guidelines [11], the gold standard of treatment for DTC and PDTC is represented by surgery (total or near-total thyroidectomy). Before cervical surgery, an accurate ultrasound evaluation of neck and mediastinum should be performed in all patients to identify lymphadenopathy that should be surgically removed simultaneously with the thyroid. Patients with intermediate or high risk of recurrent disease or with distant metastases should be treated with a subsequent ^131^I therapy and thyroid stimulating hormone (TSH) suppressive therapy [11].

For patients with metastatic DTC that progresses despite standard therapies, systemic cytotoxic chemotherapy has been evaluated both in phase 2 and retrospective studies but, to date, there is no role for its routinary use. Doxorubicin (recommended dosage: 60–75 mg/m^2^ every 3 weeks) remains the single most effective and approved cytotoxic chemotherapy for the treatment of RAI-refractory DTC patients with metastatic, rapidly progressive, symptomatic, and/or imminently threatening disease but it should be used only when cases are not manageable with other approaches. Nowadays, as described in following paragraphs, targeted therapies able to inhibit abnormal activated tyrosine kinases (TKI) represent the systemic therapies that should be used as first line in metastatic TCs.

Regarding adjuvant neck/thyroid bed/locoregional external beam radiation (EBRT), its application in DTC/PDTC patients remains controversial and, according to the literature, there is no role for routine adjuvant EBRT to the neck in DTC patients, especially when an initial complete surgical removal of the tumor is achieved. However, EBRT should be considered for those patients undergoing multiple and frequent neck re-operations for palliation of locoregional recurrences and, in some cases of bone metastases especially to control the pain that is frequently associated with these lesions. [12,13].

Recently, several studies have reported the efficacy of other localized treatments with thermal (radiofrequency or cryo-) ablation [14,15], ethanol ablation [16,17], or chemo-embolization [18] on single/few metastases or on locoregional persistence/recurrence of neoplastic disease. These kinds of treatments can be offered to those patients with high risk for surgery, those who refuse to undergo repeated surgeries, and those with oligometastatic but progressing disease. The possibility of performing these localized treatments should always be taken into consideration before the initiation of systemic treatment [11].

### 1.2. Anaplastic Thyroid Cancer

Anaplastic thyroid carcinoma (ATC) represents about 1% of all TC and is the most aggressive thyroid tumor, accounting for the majority of all TC death [19]. The ATC is a rare undifferentiated form of TC, unable to take up ^131^I and with no chance of cure [19,20]. It is associated with a rapid and lethal progression, especially at a local level. At the time of diagnosis, 30–40% of patients have locoregional metastases and/or vocal cord paralysis and 70% have direct invasion of local tissue including the trachea, muscle, esophagus, and larynx. Distant metastasis usually appears in those patients receiving aggressive treatments and can involve multiple sites including the lungs (50–80%), bone, skin, and brain (6–12%) [21]. Median survival time in ATC patients is approximately 5–6 months, and only 10–15% of patients survive 2 years after presentation [19,21]. Due to this fatal outcome, TNM classification of the American Joint Committee on Cancer provides only stage IV for this tumor, which can be subdivided into IVA when the tumor is confined to the thyroid, IVB when the tumor is present beyond the thyroid gland but confined in the neck, and IVC when distant metastases are present.

The initial management of ATC should include the evaluation of airways’ stability to establish whether an immediate intervention is necessary, and the evaluation of full resectability (R0) of primary tumor, since the only debulking of the cancer does not improve the patient’s outcome. Therefore, surgery should be of first choice when a full resection of tumor can be obtained and when no distant metastases are identified. About 2 to 4 weeks after surgery, chemoradiation should be offered to the patient, while unresectable ATC should be treated directly with radiation therapy (> 60 Gy on primary tumor) and adjuvant chemotherapy. Standard cytotoxic chemotherapy includes taxane, platinum-based drugs, and doxorubicin [22].

The same approach (radiation therapy and chemotherapy without local surgery) should also be offered to those patients with small-volume metastases while systemic therapies with or without palliative radiation therapy should be considered in patients with large-volume metastases. Nowadays, only specific TKI can be used in restricted subgroups of ATC with specific molecular alterations such as *BRAF^V600E^* mutations, *RET* or *TRK* fusions as discussed below.

### 1.3. Medullary Thyroid Cancer

Medullary thyroid cancer (MTC) originates from neural crest C cells and represents 4% of all TC. The biological behavior of MTC is more severe than that of DTC: the 10-years survival rate is approximately 50% and can be lower in patients with advanced disease at diagnosis [23]. Distant metastases are observed at presentation in 7–23% of MTC patients and are the main cause of MTC-related death. Total thyroidectomy and central compartment neck dissection is the primary surgical treatment and the only curative one for localized MTC. When a widespread regional or metastatic disease is present, repeated surgeries are not associated with a higher cure rate, and less aggressive procedures should be considered. In these cases, whenever possible, according to the extension and the sites of the disease, a local treatment should be preferred while systemic therapy should be used when the disease becomes multimetastatic and rapidly progressive [24,25]. EBRT is indicated to improve the local control of disease in case of local recurrence or locoregional lymph node metastases or as palliative therapy to reduce pain from bone metastases or to treat brain metastases [25]. Radiofrequency thermo-ablation is frequently applicable to bone, liver, and lung to treat single metastases or a single progressive and symptomatic one in the context of a stable disease [26,27]. Another quite new and promising local treatment is the conventional transarterial chemoembolization (TACE) or radioembolization (TARE) commonly used in some advanced cases of liver metastatic disease, especially when liver metastases are smaller than 3 cm and the liver involvement is less than 30% [28]. Among systemic therapy, chemotherapy shows no clinical durable advantages and benefits in MTC and for this reason is no longer indicated [25]. Over the years, several types of radionuclides have been studied for the treatment of MTC, based on its neuroendocrine origin, but despite the decrease of its serum marker (i.e., calcitonin (CT)), no results have been obtained in terms of reduction of size and number of metastatic lesions [29]. Recently, promising results have been shown for the peptide receptor radionuclide therapy using 177Lu-labeled or 90Y-labeled somatostatin analogues, but it is limited to MTC cases with significant somatostatin receptors expression [30].

Nowadays, the current systemic options for the treatment of advanced and progressive MTC are both multi-targeted and specific TKI as we will describe thereafter.

## 2. Intracellular Signaling Pathways in Thyroid Cancer

In the last decades, the mutations identified by several studies and by the Cancer Genome Atlas (TCGA) show different patterns mostly affecting two signaling pathways: mitogen-activated protein kinase (MAPK) and phosphatidylinositol-3 kinase (PI3K) pathways [31,32,33] (Figure 1). When a mutation at any level of these pathways occurs, resulting from a somatic or germline genetic alteration of the transmembrane receptors (TKR) or from a central mediator, the signaling becomes constitutively activated, causing an uncontrolled cell replication, loss of cell differentiation, and decrease of cell apoptosis. Both genetic and epigenetic modifications in pathway receptors and effectors are involved [34], and promote the progression of follicular thyroid cells to PTC or FTC. Additional mutations and/or rearrangements and an increase of the MAPK and/or PI3K pathways signaling seem to promote a further progression to PDTC, while further genetic events, especially involving *p53*, epigenetic alteration, and infiltration of immune cells, lead to the onset of ATC (Figure 2).

### 2.1. MAPK and PI3K Pathways

The most common genetic alterations in TC associated with MAPK and PI3K signaling pathways include mutations in central mediators such as *BRAF*, *RAS*, *PTEN*, *AKT*, and *PI3KCA* (Figure 2).

*BRAF^V600E^* is the most common mutation in PTC. It was described in 18–87% of PTCs and, less frequently, in PDTC, ATC, and HTC [35,36], and it is mutually exclusive with other mutations, thus suggesting that the presence of one single driver mutation is sufficient for thyroid tumorigenesis [37]. *BRAF^V600E^* mutations, as well as oncogenic activation of MEK/ERK, also play an important role in the loss of sodium iodide symporter (NIS) activity, leading to RAI refractoriness [38].

*RAS* mutations are reported in 30–40% of FTC, in 30–45% of follicular variant PTC (rarely in classical variant of PTC), in 15% of HTC, in 20–40% of PDTC and ATC, and in 10–30% of MTC [39,40,41,42].

Although their frequency is low [43], *PTEN*, *AKT*, and *PIK3CA* are the most common mutated mediators in PI3K pathways. Mutations, deletions, and epigenetic changes of oncosuppressor *PTEN* were described in DTC, PDTC, ATC, and HTC [33,35,42]. *AKT1*, *AKT2*, and *AKT3* were reported in very few cases of TC and some *AKT1* mutations were reported in ATC and in FTC [44] and in a TC metastasis, suggesting that this mutation can occur late during cancer progression [45]. As well as for *AKT1*, *PIK3CA* amplifications, and less commonly mutations, were observed late in TC progression and more frequently in ATC than DTC [45,46].

*RET* proto-oncogene is one of the most common altered oncogene coding for a TKR in TC [47]. As well as its occurrence in other altered central mediators, when a mutation occurred in *RET*, this oncogene became constitutively activated with a subsequent activation of MAPK and PI3K pathways [42]. In PTC, the most prevalent *RET* alterations are *RET/PTC* rearrangements. In particular, a *RET* fusion was found in about 50% of PTC diagnosed in atomic bomb survivors after a high radiation dose exposure and was very common, especially *RET/PTC1* and *RET/PTC3*, in childhood TC diagnosed after the Chernobyl accident [48,49].

In MTC, the most common genetic alterations are the *RET* oncogene gain of function point mutations, both in sporadic MTC cases, in which a somatic *RET* mutation is found in about 40–70% of patients, and in familiar/hereditary MTC cases, in which a germline mutation is present in 95% of kindred [50]. The most frequent *RET* point mutation detected in multiple endocrine neoplasia syndrome (MEN) type 2A occurs on codon 634, while in 95% of MEN 2B and in 75–95% of sporadic cases the most frequent alteration is *RET* mutation M918T [51,52]. Different *RET* mutations produce distinct phenotypes that can differ in terms of age of onset of MTC, aggressiveness of MTC, and association with other endocrine malignancies [51].

Neurothrophic receptor tyrosine kinase (NTRK) are a group of tyrosine kinase (TK) receptors encoded by the *NTRK1*, *NTRK2*, and *NTRK3* genes and their rearrangements were reported in PTC, PDTC, and ATC [33,53,54]. Due to a phosphorylation of their tyrosine residues, mutated receptors, encoded by mutated *NTRK*, lead to constitutive activation of MAPK, PI3K, and phospholipase C-signaling pathways [53,54], acting as oncogenic drivers.

Mutations and gene fusions involving anaplastic lymphoma kinase (*ALK*) were also reported, mainly in PDTC and ATC and less frequently in PTC [42,55,56]. In particular, in ATC two different mutations with aminoacidic changes within ALK TK domain were identified [57].

Growing evidence has revealed the importance of tumor microenvironment, composed of the extracellular matrix and stromal cells. Indeed, with relevant changes in microenvironment, tumors become able to survive, grow, invade, and give distant metastasis and the improvement of angiogenesis is the most important process reported [58,59].

### 2.2. Other Genetic Alteration in Thyroid Cancer

In about 30–35% of FTC but also in follicular variant of PTC (FVPTC), *PAX8/PPARγ* rearrangement is reported. The *PAX8/PPARγ* gene fusion results in the production of an oncoprotein, PAX8/PPAR-γ fusion protein, that is considered to act as a dominant-negative inhibitor of wild-type PPARγ and/or as a unique transcriptional activator of subsets of PPARγ-responsive and PAX8-responsive genes [60,61].

In 50–80% of ATC and in up to 35% of PDTC, mutations of oncosuppressor *TP53* are described. The same alteration can also be found in the most aggressive FTC and PTC [42,62]. Other common mutations in ATC and PDTC regard Wnt- pathways, including the gain of function mutation of *β-catenin* gene and the loss of function mutation of *Axin1* gene [63,64].

Another mutation frequently found in ATC (40–70%) and in PDTC (40%) involves *TERT* promoter. The same promoter alterations can also be found in the aggressive forms of HTC, PTC, and FTC [42,65].

### 2.3. Immune Microenvironment

The immune system is the major determinant of tumor microenvironment. Many studies have shown that the escape from immune response called “tumor immunoediting” has a role in thyroid tumor progression and that immune cells, both from adaptive (T cells) and innate system (macrophage), secreted soluble mediators, and immune checkpoints are the principal mediators in this scenario [66]. More in details, T regulatory lymphocytes are highly expressed in the tumor microenvironment, where they exert an immunosuppressive function and are associated with a poorer prognosis [67]. The CD4^+^CD25^+^T cells were demonstrated to be more expressed in PTC than in benign thyroid nodules and were associated with PTC invasiveness [68]. Cytotoxic T cells are, on the contrary, specialized in killing target cells and are involved in promoting tumor cells apoptosis and cellular proliferation inhibition [67]. In PTC, low cytotoxic T cells concentration was associated with bigger and more invasive tumors [69]. Tumor-associated macrophages (TAM) are the most abundant cells in tumor-infiltrate and their products that are modulated by cancer cells, are able to remodel extracellular matrix, facilitate angiogenesis, and promote tumor cell migration and invasion [70,71]. In TC, particularly in PTC, higher expression levels of TAM in tumor microenvironment were associated with lymph node metastases and poorer prognosis [72].

Immune checkpoints are regulators of T cells immune response through co-stimulatory or inhibitory molecules: cytotoxic T lymphocyte antigen4 (CTLA-4), programmed cell death protein 1 (PD-1), programmed cell death protein 1 ligand (PD-L1) inhibit the T cells mediated control on tumor cells proliferation. In TC, PDL-1 was differently expressed according to the hystotype, with higher expression levels in ATC (822.25) with respect to FTC (7.6%) and PTC (6.1%), with even higher expression in advanced TC [73]. In PTC, higher expression of CTLA-4 and PD-L1 was associated with *BRAF^V600E^* mutation and a lower degree of differentiation [74]. On the contrary, in most studies PD-L1 expression was almost negative in MTC [75]. Based on this data, two main immunotherapy strategies have been designed: one increasing the ability of body cells to kill tumoral ones (i.e., tumor vaccines and adoptive cells therapy) and the other one improving the ability of immune cells to suppress tumor development (i.e., immune checkpoint inhibitors).

## 3. Current Available Systemic Therapies for the Treatment of Advanced and Multimetastatic Thyroid Cancer

Over the last decades, several small molecular agents able to inhibit TK and TKR with different mechanisms have been generated and have been demonstrated to be effective anti-solid tumor and anti-leukemic agents with the name of TKIs. This first generation of TKIs are able to inhibit specific oncogene alterations but they also act against several other TKRs and for this reason they are also recognized as multikinase inhibitors (MKI). Sometimes, the activity of these MKI against these other receptors is stronger than the activity they have against the driver oncogene alteration of TC, and several adverse events (AEs) are due to these “off-targeted” activities.

Four MKI, lenvatinib and sorafenib for the treatment of advanced RAI-refractory DTC, and vandetanib and cabozantinib for the treatment of MTC, have lastly been approved by both the Food and Drug Administration (FDA) and the European Medical Agency (EMA).

Recently, a second generation of TKIs has been developed which are specific for a specific altered oncogene. The high specificity of these drugs is the reason for a greater tolerability because of a much smaller number of AEs having very few “off-targeted” activities.

So far, four of them, larotrectinib, entrectinib, selpercatinib, and pralsetinib have been approved by FDA and are under evaluation of EMA.

### 3.1. Systemic Treatment in DTC and PDTC: Multikinase Inhibitor Approved Drugs

Sorafenib is a small oral molecule with a strong anti-angiogenic activity. Sorafenib is able to inhibit VEGF1–3, PDGF, FGF, KIT, RET receptors and, weakly, RAF (Table 1). After two different phase 2 trials [76,77] that demonstrated its efficacy in DTC patients, a randomized, double-blinded, placebo-controlled phase 3 trial (DECISION study) [78] was designed. In DECISION study, 207 patients were randomized to the drug and received a starting dose of 400 mg twice a day, while 209 patients were randomized to placebo. A crossover from placebo to drug was available when a disease progression was demonstrated in placebo patients after they were unblinded. All 417 patients were TKI naïve. The primary endpoint was the evaluation of progression-free survival (PFS) that was significantly longer in sorafenib-group than in placebo-group (10.8 months and 5.8 months, respectively; *p* < 0.0001) with a response rate of 12.2% in the first group and of 0.5% in the second one. OS was not different in the two groups, but this was probably due to the crossover from placebo to sorafenib. Table 2 shows the most common side effects of the drug.

Initially approved for renal and hepatocellular carcinoma, DECISION study led to the approval of sorafenib for the treatment of ^131^I refractory DTC.

Since the response rate appeared to be low (12.2%) when sorafenib was used as a single-agent, a phase 2 study [79] was started to determine whether adding temsirolimus, an inhibitor of mammalian targeted of rapamycin (mTOR), to sorafenib could improve these results. Thirty-six patients with metastatic, RAI-refractory TC of follicular origin received treatment with the combination of oral sorafenib (200 mg twice daily) and intravenous temsirolimus (25 mg weekly). The best response was a partial response (PR) in 58%, and progressive disease (PD) in 3% (six patients were not evaluable for a response). Response rate was 10% in patients who had received any prior systemic treatment and 38% in those who had not received prior systemic treatment. One of two patients with ATC had an objective response. PFS at 1 year was 30.5%. The most common grade 3 and 4 toxicities were hyperglycemia, fatigue, anemia, and oral mucositis.

Sorafenib was also administered in a phase 2 study [80] in patients with ATC on the basis that the *BRAF* oncogene is mutated to its active form in up to 24% of ATC cases. Twenty ATC patients were enrolled and sorafenib was administered at 400 mg twice daily. The evaluation of the disease by RECIST revealed 10% of PR and 25% of stable disease (SD). The duration of response in the two responders was 10 and 27 months, respectively. For the patients with SD, the median duration was 4 months (range 3–11 months). The overall median PFS was 1.9 months with a median and a 1-year survival of 3.9 months and 20%, respectively. Furthermore, in this study, toxicity was manageable and as previously described for sorafenib, included hypertension and skin rash. The authors concluded that, although sorafenib has activity in ATC, this is at a low frequency and is similar to the previous experience with fosbretabulin [81].

Lenvatinib, as well as sorafenib, is an oral TKI with a high anti-angiogenic activity. Lenvatinib is able to inhibit VEGF1–3, FGF1–4, PDGF, KIT, and RET receptors (Table 1). The efficacy of lenvatinib in TC, both DTC and MTC, was first demonstrated in a phase 2 study [82], and, then, a multicentric, randomized, placebo-controlled phase 3 study was designed (SELECT study) [83]. A total of 392 patients were enrolled in the study and, in particular, 261 were randomized in lenvatinib arm, starting with 24 mg/day of drug, and 131 patients in placebo arm. Patients who received, before the enrolment, a previous treatment with no more than one TKI were allowed. The primary endpoint was the evaluation of PFS that was significantly longer in lenvatinib-group than in placebo-group (18.3 months and 3.6 months, respectively; *p* < 0.001) with 4 complete responses (CR) in lenvatinib-group. A sub-analysis confirmed that PFS was also longer in lenvatinib-group when patients receiving a previous TKI therapy were considered. Although OS was not different in the two groups, when a sub-analysis stratified the patients by age, an OS advantage in patients older than 65 years old was demonstrated. Table 2 shows the most common side effects of this drug. No unexpected toxicities were reported.

A post-marketing multicenter, randomized, double-blind phase 2 study was conducted to evaluate if a lower starting dose could demonstrate the same clinical benefits with a lower AEs incidence (Study 311, NCT02702388). The study is still active but is no longer recruiting and 152 participants have been enrolled. They were randomly assigned to treatment with 1 of 2 blinded dosages of lenvatinib in a 1:1 ratio to receive lenvatinib 18 mg or 24 mg orally once daily. The estimated study completion date was September 2020 and, even if the full results have not yet been presented, in a press release from EISAI company topline the results were shown. According to these results, the 18 mg dose did not show non-inferiority in efficacy in respect with the 24 mg dose as measured by ORR at week 24, and the incidences of grade 3 or higher treatment-emergent AEs through week 24 were similar between dose arms.

Before and after the commercialization of the drug, several studies were published about the real-life use of the drug showing the same safety results with a lower PFS [84,85]. An expanded access program (EAP) was also initiated in Italy to evaluate the drug’s safety compared to the SELECT study and other series and patients’ quality of life [86]. In this study, all AEs that occurred during the 6 months of lenvatinib treatment in 39 RAI-DTC patients were recorded. According to these results, the safety profile of lenvatinib was similar to that already reported in other studies with differences in percentages and grading: a lower percentage of patients experienced diarrhea (36.1%), weight loss (30.5%), nausea (11.1%), and proteinuria (11.15%). Regarding the evaluation of quality of life, the EAP study showed a trend of improvement of the general health status and a reduction of symptoms correlated to the disease.

Lenvatinib was also demonstrated to be effective in a retrospective, single center analysis of 18 Korean patients with confirmed ATC [87]. Six patients had resectable disease that progressed after a combination of surgery, radiotherapy, and chemotherapy, and 12 had unresectable disease that progressed after radiation treatment and chemotherapy. Median OS for the 18 lenvatinib-treated patients was 230 days while the survival rates at 6 months and 1 year were 61.1% and 22.2%, respectively. Three patients survived beyond 1 year; 15 patients died, of whom four had local disease and 11 had distant metastasis. Two patients had tumor volume increases of 9–10%. The other 16 patients had tumor volume reductions of 2–69%. Six patients had tumor volume reduction ≥ 50%. Similar results in ATC patients were obtained in a previous study of Takahashi et al. [88].

To date, lenvatinib is approved for the treatment of RAI refractory DTC, hepatocellular carcinoma, and, in combination with everolimus, for renal cell carcinoma, but not for ATC.

### 3.2. Systemic Treatment in MTC: Multikinase Inhibitor Approved Drugs

Vandetanib is a once-daily oral multitargeted drug with a recommended daily dose of 300 mg/die, additionally adjusted in case of unmanageable toxicity. Its main molecular targets are represented by VEGFR-2, VEGFR3, EGFR, KIT, and RET as shown in Table 1. Vandetanib was the first TKI approved for the treatment of symptomatic, unresectable, locally advanced or metastatic MTC in adult patients, both by FDA (2011) and EMA (2013). The approval of the drug was due to the results of the international, multicentric and randomized against placebo phase 3 trial (ZETA study) showing a significant longer median progression-free survival in treated patients with respect to the placebo ones (30.5 months versus 19.3 months) [89]. Moreover, a significant clinical benefit in treated patients was also demonstrated for objective response rate (ORR) (*p* < 0.001), disease control rate (*p* < 0.001), and biochemical response (*p* < 0.001). The use of this drug was also approved for children with advanced hereditary MTC thanks to the results obtained in a phase 2 clinical trial [90].

The treatment was generally well tolerated, and its side effect profile was similar to that of other TKIs. Table 2 shows the most frequent AEs at any grade [89]. The QT elongation is a drug-specific side effect and potentially life-threatening, occurring in 14% of treated patients [89].

With the daily use of vandetanib in real life, some retrospective studies were published in order to find predictive factors for a more durable drug response [91,92]: a younger age at the disease diagnosis and enrolment was able to predict a longer and more durable response to vandetanib. A positive correlation was also found between the presence of AEs, any type, and a better response to treatment [91].

Cabozantinib is an oral multiple TKI approved by the FDA in 2012 and EMA in 2014 for the treatment of progressive and metastatic MTC. It is able to inhibit three main TK receptors: MET, VEGFR2, and RET as shown in Table 1. In the phase 3 clinical trial (EXAM study) a significantly longer median PFS was observed in patients treated with the drug compared to those treated with placebo (11.2 versus 4.0 months) and also a significant difference in the overall response rate corresponding to 28 versus 0%, respectively [93]. Despite these good results, also cabozantinib was not able to improve the OS, except for the subgroup of patients with M918T-*RET* mutation which showed a significant increase of OS (median survival 44.3 months compared to 18.9 months) [94]. Moreover, cabozantinib seems to also be effective as second-line therapy: in fact, no differences for PFS were observed in the subgroups of naive and previously treated MTC patients and this data could be useful in the real-life management of the drug.

The most frequent AEs occurring at any grade in the EXAM study are reported in Table 2 [93]. Fatal adverse reactions occurred in 6% of patients receiving cabozantinib and the death causes were hemorrhage, pneumonia, septicemia, fistulas, cardiac arrest, and respiratory failure. Interestingly, while hemorrhage was present both in the cabozantinib and placebo arm, gastrointestinal perforations (3%) and fistulas (4%) were diagnosed exclusively in the cabozantinib arm. No cardiological AEs were observed, in particular no QT elongation was reported.

Another clinical trial with cabozantinib in MTC patients has been conducted and nowadays is still active but no longer recruiting. The randomized, double blind, multicentric EXAMINER clinical study (NCT01896479) has been designed to evaluate two different doses of cabozantinib (60 mg vs. 140 mg) in progressive and metastatic MTC in order to test if the lower dose has the same results in terms of PFS and overall response rate compared to higher dose but with fewer AEs. According to the study design, previously other TKI treatment was permitted, and patients have been stratified based on M918T*-RET* status and after randomization they have been treated until disease progression or intolerable toxicity. The study is still ongoing, and no data have been presented so far.

### 3.3. Systemic Treatment in DTC, PDTC, and ATC: Specific Inhibitors of NTRK-Fusions

NTRK fusions are rare but can be found in several solid tumors, including PTC, PDTC, and ATC. When present in TC, this fusion is usually mutually exclusive with other oncogenic driver mutations. Larotrectinib is a highly selective inhibitor of tropomyosin receptor kinase (TRK) A, B, and C (Table 1), and is approved by FDA and EMA for the treatment of all solid cancers that harbor NTRK fusions, both in adult and in pediatric patients. The results of two phase 1 trials and of one phase 2 trial were all together reported in a primary analysis [95] and then in an additional paper with 67 supplemental patients [96]. Supplemental analysis included 109 patients with solid tumors, comprising 19% of TC. Sixty-three percent of patients had PR (including 10 TC patients) and 17% a CR (including 3 patients with TC). Recently, an ORR of 79% and a median duration of response of 35.2 months were reported in patients with TRK fusion cancer in various tumor types treated with larotrectinib [97]. Moreover, at 2020 ESMO meeting, Cabanillas et al. presented the preliminary data on efficacy and safety of larotrectinib in adult and pediatric patients with TRK fusion TC [98]. A total of 28 TC patients (19 PTC, 7 ATC, and 2 FTC) with TRK fusion (43% NTRK1 fusion and 57% NTRK3 fusion) were included. ORR was 75%, including 29% for patients with ATC, and the duration of response ranged from 1.9 to 41.0 months. Median PFS was not reached and the 12-month PFS rate was 81%. AEs were mostly grade 1–2 and only 7% of patients had grade ≥3 AEs related to larotrectinib, demonstrating a favorable safety profile. The most common AEs are summarized in Table 2. Interestingly, although no preclinical in vitro thyroid model to assess the capacity of NTRK inhibitors to promote redifferentiation is present, a case report was published showing that larotrectinib may restore RAI uptake, likely by inhibiting signaling pathways activated by *EML4-NTRK3* in a manner similar to the action of MAPK pathway inhibitors [99].

Entrectinib is another selective inhibitor of TRKA, TRKB, and TRKC, but it is also able to inhibit ALK and ROS1 TK (Table 1). A recently integrated analysis of three phase 1–2 trials of entrectinib in patients with advanced or metastatic NTRK fusion positive solid tumors [100] showed that 7% of patients had a CR and 50% a PR with a manageable safety profile. Since the peculiarity of this drug is its capability to penetrate the blood–brain barrier, targeting brain metastasis or primary brain tumors, an updated integrated analysis of the above-mentioned entrectinib studies, focusing on intracranial activity of entrectinib, was presented at 2020 ESMO meeting [101]. According to these authors, the ORR was 50% and the intracranial efficacy was evident regardless of prior brain radiotherapy. The most common AEs of entrectinib, most of grade 1 or 2, are reported in Table 2.

### 3.4. Systemic Treatment in DTC, PDTC, ATC, and MTC: Specific Inhibitors of RET Oncogene Alterations

Selpercatinib, known also as LOXO-292, is a highly selective ATP-competitive small molecule *RET* inhibitor (Table 1), with a half-life of 32 h. As demonstrated in experimental models, this new compound has a very high potency to inhibit different *RET* alterations, both point mutations and fusions, including the V804M mutation responsible for other TKI resistance [102] and shows an antitumor activity in brain. At variance, its activity against the other receptors, such as VEGFRs, is very low, making the drug highly specific for *RET* alterations. From May 2017, a phase 1/2 single-arm, multicenter, open-label, multi-cohort clinical trial (LIBRETTO-001 trial, NCT03157128) has been conducted in 65 centers in 12 countries to evaluate the efficacy of LOXO-292 in solid tumors with RET alterations (*RET* mutations and *RET* fusions) among with MTC and TC derived from follicular cells (PTC, PDTC, and ATC) [103]. The primary end point was the evaluation of the objective response; whole secondary ones included the duration of response, the PFS, and the safety. The drug was administered orally in 28 day-cycles and the study was conducted in 2 parts, phase 1 (dose escalation) and phase 2 (dose expansion): patients enrolled in the phase 1 dose-escalation group received drug doses ranging from 20 mg once daily to 240 mg twice daily, while patients enrolled in the phase 2 dose escalation group received 160 mg twice daily. The data cut-off date was December 16, 2019. A total of 143 MTC patients were treated across different groups: 55 patients with *RET* mutations, previously treated with vandetanib or cabozantinib or both, and 88 patients with *RET* mutations not previously treated with other TKI. Selpercatinib showed a marked and durable anti-tumor activity. In fact, the percentage of patients experiencing a response was 69% (95% CI, 55 to 81) in the first cohort and 73% (95% CI, 62 to 82) in the second one and interestingly, the first cohort was independent by the number of previous multitargeted treatments administered. The 1-year progression-free survival was 82% (95% CI, 69 to 90) in the first group and 92% (95% CI, 82 to 97) in the second one. Responses were observed across all qualifying *RET* mutations, including in patients who had tumors harboring the drug resistant *RET* mutation V804.

Although the majority of patients enrolled had MTC, 19 DTC patients (13 PTC, 2 PDTC, and 1 HTC) and 2 ATC patients were also included [103]. In 19 patients with previously treated *RET* fusion–positive DTC, the percentage who had an objective response was 79%, and 1-year PFS was 64%. Of two patients with ATC, one had a durable response for 18 months, with the response ongoing at the end of the study.

The most related AEs were of grade 1–2. Most frequent any grade AEs are reported in Table 2. The 2% of all patients discontinued selpercatinib for drug-related AEs, the most common of which were an increased alanine aminotransferase level and drug hypersensitivity. The study is still ongoing and a longer follow-up of the patients involved in this trial will be needed to define the ultimate durability of Selpercatinib efficacy across all cohorts. On the basis of these results, on 8 May 2020, the FDA granted accelerated approval to selpercatinib among other (i.e., non-small cells lung cancers [NSCLC]) for adult and pediatric patients ≥ 12 years of age with advanced or metastatic *RET*-mutant TC, both MTC and DTC/ATC, who require systemic therapy [104]. The approved dose was 160 mg twice daily for patients with body weight ≥ 50 kg and 120 mg twice daily for patients with a lower body weight.

An interventional phase 3 trial with selpercatinib has recently started and is recruiting all over the world. This is a multicenter, randomized, open-label study comparing selpercatinib to physicians’ choice of cabozantinib or vandetanib in patients with progressive, advanced, kinase inhibitor naïve, *RET*-mutant MTC (LIBRETTO-531 trial, NCT04211337) [105]. Patients will be randomized with a 2:1 ratio to two different arms: arm A only selpercatinib and arm B cabozantinib or vandetanib at physician decision, with the possibility of crossover to selpercatinib at progression. Treatment failure-free survival, including radiographic progressive disease, unacceptable toxicity (predefined by protocol), or death from any cause, is the primary endpoint. Secondary end points are the PFS, overall response rate, duration of response, OS, PFS 2 by investigator (from baseline to second disease progression or death from any cause), ORR by *RET* mutation status, safety profile, and pharmacokinetics.

Currently, a few other phase 1–2 studies are ongoing, aiming to confirm the efficacy and tolerability of selpercatinib in different types of solid tumors with *RET*-alterations: an expanded access trial (NCT03906331) for patients with TC with *RET* activation who have been previously treated with vandetanib or cabozantinib or other TKI and cannot enter in the phase 3 trial and a phase 2 trial dedicated exclusively to Chinese patients (NCT04280081).

Pralsetinib, also known as BLU-667, is another highly selective *RET* inhibitor targeting oncogenic *RET* alterations (Table 1). Previous in vitro studies demonstrated its almost 10-fold higher potency over other MKIs against oncogenic *RET* variants and resistance mutants while in vivo studies showed that the drug is able to inhibit the growth of NSCLC and TC xenografts driven by various *RET* mutations and fusions without inhibiting VEGFR2 [106]. In the ARROW clinical study (NCT03037385), a phase 1/2 open-label multicenter trial, the drug was studied in patients with *RET*-mutated TC and other solid tumors. The study was conducted at 75 sites in 11 countries, and consisted of a phase 1 dose escalation, already completed, with administered dose of 30–600 mg daily, that determined the recommended phase 2 pralsetinib dose as 400 mg orally once daily, and a phase 2 expansion cohort defined by tumor type and/or *RET* alteration in patients treated with 400 mg once daily. The primary objectives in phase 2 included overall response rate and duration of response and safety. At the ESMO Virtual Congress 2020, some preliminary data were shown [107]. At 13 February 2020, in the 79 patients with *RET* mutation positive MTC the overall response rate was 65% (51 of 79), in particular it was 74% in treatment-naïve patients and 60% in those previously treated with cabozantinib or vandetanib. The 18-months PFS and duration of response were 71% and 90% in patients previously treated with other TKI while in treatment-naive patients they were 85% and 86%, respectively. Responses occurred regardless of *RET* genotypes.

Preliminary data showed that pralsetinib was also highly active in patients with TC with *RET* fusions which showed an ORR of 75% with a median duration of response of 14.5 months.

Regarding pralsetinib’s safety, most treatment-related AEs were grade 1–2. The most common of any grade AEs are summarized in Table 2. Treatment-related AEs accounted for 4% of treatment discontinuations. On the basis of these results, pralsetinib has been approved by FDA both for the treatment of *RET*-fusion positive NSCLC and, very recently, for *RET* fusion-positive TC and *RET* mutation-positive MTC. Currently, it is under regulatory review in Europe for *RET* fusion-positive NSCLC.

### 3.5. Systemic Treatment in DTC, PDTC, and ATC: Specific Inhibitors of BRAF^V600E^ Mutation

Vemurafenib is an oncogenic *BRAF* kinase inhibitor (Table 1) approved for *BRAF*-positive melanoma with a clinical benefit in three patients with *BRAF^V600E^*positive PTC in a phase 1 trial.

The results of an open-label, non-randomized, phase 2 multicentric trial confirmed the efficacy and safety of vemurafenib in patients with histologically confirmed recurrent or metastatic RAI-refractory PTC and who are positive for the *BRAF*^V600E^ mutation [108]. In particular, PR was recorded in 10/26 patients that had never received a multikinase inhibitor. The most common grade 3 and 4 AEs were squamous cell carcinoma of the skin, lymphopenia, and increased γ-glutamyltransferase. Serious AEs occurred in 62% of patients which had never received a prior TKI and in 68% of patients which previously received a TKI. Efficacy of vemurafenib in ATC was not well established since only seven patients were evaluated in a phase 2 study [109] that involved several multiple non-melanoma cancers with *BRAF^V600E^* mutations. Of these seven patients, only one showed a CR and one a PR. To date, no phase 3 study has been designed for vemurafenib in TC.

Dabrafenib is a selective inhibitor of *BRAF^V600E^* kinase (Table 1). FDA approved the use of dabrafenib in combination with trametinib, an inhibitor of MEK1–2 kinase, for the treatment of ATC, melanoma, and NSCLC, harboring *BRAF^V600E^* mutation. The approval of these two drugs for the treatment of ATC was based on the results of a phase 2 study published in 2018 [110]. In this trial, 16 patients with *BRAF^V600E^*-mutated ATC were evaluable, after they had received prior radiation treatment and/or surgery, and six had received prior systemic therapy. The confirmed ORR was 69% with seven ongoing responses at the end of the study. Median duration of response, PFS, and OS were not reached as a result of a lack of events, with 12-month estimates of 90%, 79%, and 80%, respectively. The safety population was composed of 100 patients who were enrolled with seven rare tumor histologies. Common AEs were fatigue (38%), pyrexia (37%), and nausea (35%). In 2018, in ESMO meeting, Keam et al. proposed an updated report of the above-mentioned trial, including 28 patients [111]. According to this author, the median PFS was 60 weeks, and the median OS was 86 weeks. Dabrafenib alone and dabrafenib in combination with trametinib were also evaluated in *BRAF^V600E^* mutated RAI-refractory PTC who had evidence of disease progression within 13 months prior to randomization [112]. This phase 2 trial included 53 patients, and, in particular, 26 patients received dabrafenib (arm A) and 27 patients received the combination treatment (arm B). PR was reported in 38% of patients in arm A and in 33% of patients of arm B while an ORR, defined as a 20% to 29% decrease in the sum of targeted lesions, was 50% and 54%, respectively.

## 4. Drugs under Evaluation

### 4.1. Ongoing Studies on Follicular-Derived TC

Cabozantinib, as mentioned above, is an oral TKI approved for the treatment of advanced MTC but its efficacy was also tested in phase 1 [113] and phase 2 trials [114] in RAI-refractory patients. In the single-arm, open-label phase 1 study, 15 patients with metastatic or surgically unresectable RAI-refractory DTC were enrolled, receiving 140 mg oral daily dose of cabozantinib. Median PFS and OS were not reached but a PR was reported in 53% of patients. The study showed that the most common AEs were diarrhea, nausea, fatigue, and decreased appetite. Since an objective response to cabozantinib in phase 1 study was achieved in 5/8 patients with DTC previously treated with a VEGFR-targeted therapy, in phase 2 trial the attention was focused on cabozantinib as second or third line of treatment. In particular, only patients with a documented progression of the disease on previous TKI treatment were enrolled (*n* = 25). Twenty-one patients had received only one prior VEGFR-targeted therapy (sorafenib, pazopanib, or cediranib), and four patients had received two of such therapies. Of the 25 patients, 40% had a PR, 52% SD, and 8% had non-evaluable disease. The median PFS and OS were 12.7 months and 34.7 months, respectively, demonstrating that cabozantinib had a clinically significant, durable objective response activity in patients with RAI-refractory DTC who experienced disease progression while taking prior VEGFR-targeted therapy.

A phase 3, multicenter, randomized, double-blind, placebo-controlled study of cabozantinib in subjects with RAI-refractory DTC after prior VEGFR-TKI therapy is currently ongoing (COSMIC-311 trial; NCT03690388). The study provides for the enrollment of 300 patients. Estimated study completion date is December 2022.

Apatinib, an oral inhibitor of VEGFR2, showed clinical activity in preliminary studies involving patients with RAI-refractory DTC. These first results led to a randomized, double-blind, multicenter phase 3 trial that aimed to evaluate the efficacy and safety of apatinib in 92 patients with RAI-refractory DTC (NCT03048877). The results of the study were recently presented at ESMO meeting 2020 by Lin et al. [115] and they confirmed the efficacy and safety of this drug. Forty-six patients were randomized to apatinib arm and 46 to placebo arm. The median PFS was 22.21 months in apatinib group, and 4.47 months in the placebo group (*p* < 0.0001). ORR was 55.56% and 2.27%, respectively, while median OS was 29.9 months in the placebo arm, and not reached in apatinib arm (*p* = 0.0356). The most frequent treatment-emergent ≥ grade 3 AEs in two arms were hypertension (34.8% vs. 0%), hand–foot syndrome (17.4% vs. 0%), and proteinuria (17.4% vs. 2.2%).

In the last few years, the attention was focused on another TKR inhibitor, anlotinib (AL3818), a novel oral drug targeting VEGFR2 and 3, FGF 1–4, PDGFR α and β, c-Kit, and *RET*. Its antitumor effects were demonstrated in various types of carcinoma in a phase 1 clinical trial [116], and, recently, by Ruan et al. [117], both in vitro and in vivo, in advanced TC. Efficacy and safety of anlotinib were also confirmed by a randomized, double-blind, placebo-controlled, multicenter phase 2 trial (NCT02586337) in patients with measurable, pathologically confirmed, locally advanced or metastatic RAI-refractory DTC. The results of this study were recently presented at ESMO meeting 2020 [118]. A total of 113 patients (76 in anlotinib arm and 37 in placebo arm) were enrolled, receiving anlotinib or placebo with a dose of 12 mg QD for 2 weeks followed by a week of rest (2/1 schedule). The median PFS was 40.54 months in anlotinib arm and 8.38 months in placebo arm (*p* < 0.0001). The ORR was 59.21% in anlotinib arm and no response was observed in placebo arm (*p* < 0.0001). In addition, significant disease control rate benefit was observed for anlotinib treatment (anlotinib arm vs. placebo arm = 97.37% vs. 78.38%, *p* = 0.002). The incidence of treatment-related AEs of two groups was 100% and 86.49% (*p* = 0.003). Serious treatment-related AEs occurred in 15.79% of patients receiving anlotinib. The most common AEs in anlotinib arm were hypertension (84.21%) and hypertriglyceridemia (68.42%).

### 4.2. Redifferentiation Treatment in DTC

In the past, several drugs have been evaluated to redifferentiate metastatic TC through restoration of the NIS activity (i.e., retinoids [119], lithium [120]), but a very modest clinical benefit was found. To date, for redifferentiation in DTC, only *MEK* and *BRAF* inhibitors showed some positive results.

Selumetinib is a *MEK1/2* inhibitor and it was demonstrated to enhance RAI uptake in DTC [121]. In the study of Ho et al. [121], 20 patients with RAI-refractory DTC received 75 mg/daily of selumetinib and, although 12 of them showed an increased uptake of RAI, only eight reached an adequate threshold for the treatment. According to this study, 8/8 of RAI-treated patients obtained an objective response. *RAS* positive cases were apparently more responsive. Based on these results, a phase 2 trial with a similar design was started in the USA (NCT02393690). It is still active but with no more recruiting and with no preliminary available results.

A phase 3 study (NCT01843062) evaluating the CR rate for selumetinib in the setting of adjuvant treatment with RAI has been recently reported [122]. The results demonstrated that selumetinib combined with RAI did not improve CR rate in patients with high risk of primary treatment failure. So far, the possibility of using selumetinib in clinical practice is still under evaluation.

As mentioned above, vemurafenib is an oncogenic *BRAF* kinase inhibitor and it was tested in a pilot study aimed at evaluating the proportion of patients in whom vemurafenib increased RAI incorporation [123]. Twelve *BRAF* mutated cases were enrolled but only four patients were ^124^iodine responders on vemurafenib and treated with RAI. A tumor regression was observed 6 months after the treatment. Analysis of research of tumor biopsies demonstrated that vemurafenib inhibition of the MAPK pathway was associated with increased thyroid gene expression and RAI uptake.

Dabrafenib is another selective *BRAF* inhibitor tested to determine whether it can stimulate RAI uptake in *BRAF^V600E^* mutated unresectable or metastatic RAI-refractory PTC [124]. Ten patients were enrolled and each patient received 150 mg of dabrafenib twice daily for 25 days before TSH stimulated ^131^I whole body scan. Patients whose scan showed new sites of RAI uptake remained on dabrafenib for a longer time, and then were treated with 150 mCi (5.5 GBq) of RAI. Sixty percent of patients demonstrated sites of RAI uptake and all showed a clinical benefit, either PR or SD, on standard radiographic restaging at 3 months. Serum thyroglobulin decreased in 4/6 treated patients.

Although other studies confirmed that targeted therapy may downregulate MAPK signaling and sensitize tumors to RAI [125], so far they did not reach the clinical practice. Additional studies with a greater number of enrolled patients are required to define AEs, response duration, and survival impact of these drugs.

### 4.3. Ongoing Studies on MTC

Ponatinib (AP24534) is an oral multitargeted TKI that potently inhibits several targets, among which are both the native and mutant forms of BCR-ABL chimeric protein. Quite recently, it has been demonstrated that the drug is also a potent inhibitor of *RET* kinases and has promising preclinical activity in models of *RET*-driven MTC [126,127]. A phase 2 open label study was designed to study the efficacy and safety of this compound in advanced or metastatic MTC (NCT03838692). The primary end point of this study is to determine the objective overall response rate to ponatinib, both in patients previously treated with cabozantinib or vandetanib and with or without *RET* mutated tumors. The drug was administered at a daily dose of 45 mg. The estimated study completion is June 2021.

Anlotinib (AL3818) is a novel multi-targeted TKI, inhibiting tumor angiogenesis and proliferative signaling. After the results of a phase 2 single-arm trial (NCT01874873) demonstrating that anlotinib has a durable antitumor activity with a manageable adverse event profile in locally advanced or metastatic MTC, a subsequent randomized, double-blind, placebo-controlled, multicenter clinical trial was conducted in China to compare the efficacy and safety of anlotinib versus placebo in MTC patients (ALTER01031 trial, NCT02586350). The primary endpoint was the PFS. Previous TKI therapies were not allowed. At the 2019 ASCO congress, some results were presented [71]: 91 patients were randomized: 62 to anlotinib arm and 29 to placebo arm and median PFS was 20.67 months (95% CI, 14.03–34.63) in anlotinib arm vs. 11.07 months (95% CI, 5.82–14.32) in placebo arm (HR 0.53, *p* = 0.0289). The most common AEs in anlotinib arm were hand–foot syndrome, hypertension, hypertriglyceridemia, and diarrhea.

TPX-0046 is a 3rd generation, highly selective TKI tested, so far, in drug-resistant and naïve *RET*-driven cancer models [128]. This drug is very promising, especially for cases in which a resistance to the other TKIs has been developed through the induction of *RET-*810 mutation. Hopefully, in vivo clinical trials will confirm these in vitro studies.

### 4.4. Immunotherapy: The Present and the Future

To date, no immunotherapy has been approved for advanced TC, but several preclinical and clinical trials together with case reports have been reported. A few studies are ongoing, using immune checkpoint inhibitors alone or in combined strategy.

In the phase Ib KEYNOTE trial, the use of pembrolizumab (anti PD-1) in monotherapy was studied in TC including PTC and FTC showing an ORR of 9% and reduction of tumor size of 35–50% [129]. The use of another anti PD-1 agent in monotherapy (spartalizumab) was evaluated in progressive ATC and an ORR of 19% was observed together with a partial and complete radiological response in five and three patients, respectively [130]. In a randomized phase 2 clinical trial (NCT03246958), still ongoing but no longer recruiting, the efficacy of the combination of nivolumab (anti PD-1) and ipilimumab (anti CTLA-4) was evaluated in RAI refractory DTC, ATC, and MTC.

Considering the slow onset of immunotherapy efficacy and the evidence that immune checkpoint inhibition alone is not curative in most patients, several combined approaches with chemo and target therapy are under investigation. For example, the association of encorafenib (*BRAF* inhibitor), binimetinib (*MEK* inhibitor), and nivolumab is under evaluation in *BRAF^V600E^* mutated TC in a randomized open label clinical trial still recruiting (NCT04061980). Combined approaches with radio and chemotherapy were also evaluated in ATC: the association of pembrolizumab and chemoradiation [131] (doxorubicin or docetaxel+ Volumetric Modulated Arc Therapy) showed favorable but also limited results due to toxicity as well as the combination of durvalumab (anti PD-L1), tremelimubab (anti CTLA-4), and radio or chemotherapy [132].

Based on the evidence that VEGF inhibition can reverse the immune suppression in the tumoral microenvironment, multiple clinical trials with VEGF and/or VEGF inhibitor and immune checkpoint inhibitors have been designed. In an ongoing phase 2 clinical trial, the use of pemproblizumab+lenvatinib was investigated in unresectable ATC (NCT04171622) and also in a randomized study in a small group of advanced ATC and PDTC [133] with an ORR of 75%. Moreover, the same combination is under study in DTC and PDTC (NCT02973997) in naïve or progressing after lenvatinib patients. A triple combined therapy is under evaluation for DTC and PDTC using cabozantinib plus nivolumab and ipilimumab (NCT03914300). A comparison of different combined approaches based on mutational status of ATC and PDTC cancer is ongoing at the MD Anderson Cancer Center: all patients received atezolizumab plus other treatments according to the stratification in four groups of therapy based on the presence of *BRAF* mutation (vemurafenib plus cobimetinib), *RAS*, or NIF1/2 (only combimetinib) or the absence (bevacizumab plus paclitaxel) (NCT03181100).

Few studies are ongoing for MTC. A phase 2 trial of pembrolizumab is ongoing in recurrent or metastatic MTC (NCT03072160) as well as the use of a cancer vaccine GI-6207 targeting the CEA in a randomized phase 2 clinical trial (NCT01856920). A future prospect for MTC patients might be the combination of specific *RET* inhibitors (LOXO292 and BLU-667) with immune checkpoint inhibitors.

So far, immunotherapy is not yet applicable in clinical practice for patients with advanced TC but, as demonstrated by the numerous active trials, there is great interest and hope.

## 5. Open Issues

Despite good results, both from clinical trials and clinical practice, some problems are still present in the use of these therapies. For example, problems are the balance between drug efficacy and management of AEs, the drug resistance, and the absence of recognized predictors of response. While for AEs, thanks to the daily use of these drugs in clinical practice, some preventive strategies have been implemented to avoid toxicity and to prolong the use and efficacy of the treatment [134], the questions regarding the escape phenomenon and the finding of predictive markers of response are still open.

The availability of predictive markers would help to select patients with higher probability to benefit from the treatment, but the research into these genetic and/or biological markers is still ongoing. The pre-treatment analysis of the mutational pattern of tumor samples might have a role in the selection of patients, but, in almost all clinical trials, the drug response was demonstrated to not be strictly related to the mutational status of tumors with some exceptions. For examples, for cabozantinib the presence of M918T somatic *RET* mutation in tumor sample seems to be associated with better overall survival [94], while on the contrary for vandetanib a drug resistance was associated with V804M *RET* mutation [102]. Furthermore, the occurrence of same AEs has been proposed as a predictive response marker of drug efficacy. For example, hypertension was significantly correlated with improved outcomes during lenvatinib treatment [135], while the occurrence of any type of AEs was associated with long-term durable response for vandetanib [91]. The same was observed for skin toxicity during sorafenib treatment in hepatocellular carcinoma [136], for hypertension, hypothyroidism, and diarrhea for cabozantinib in renal cancer [137], and proteinuria in MTC [138]. In addition, the expression profiling of miRNAs in tumor tissue and/or circulating cytokine/angiogenic factors in blood samples are under study. For example, baseline higher levels of fibroblast growth factor 23 and angiopoietin-2 were demonstrated to be predictive of a better progression-free survival in DTC treated with lenvatinib [139]. The same was demonstrated for higher expression levels in tumor samples of FLT1, FLT3, and VEGFB for vandetanib treatment in sporadic MTC [140].

Drug resistance is one of the most critical and unavoidable problems for the clinician. Many mechanisms limit drug efficacy but especially the tumor heterogenicity, which is related not only to a genetic intratumor heterogenicity but also to epigenetic alterations induced by the molecules secreted by the tumor microenvironment. Much evidence shows that a genetic heterogenicity is present in TC and related to different subclones within the same tumor [141]. This heterogenicity can evolve during tumor progression or as a consequence of a drug-dependent selection of a pre-existing or newly acquired resistant clone [142]. Other than this mechanism, the crosstalk between tumor microenvironment cells (T cell, fibroblast, macrophages) and cancer cells can also alter tumor phenotype by increasing tumor plasticity and its stem potential [143]. The use of “liquid biopsies” and search of circulating DNA seems to be an important emergent strategy to monitor both the treatment efficacy and the development of new mutations able to induce drug resistance [144].

## 6. Conclusions

Thyroid cancer has different histologies and molecular identities. The initial treatment is always represented by total thyroidectomy plus/minus lymphadenectomy followed by, whenever appropriate, RAI treatment. This initial treatment can cure the vast majority of TC but the subgroups of multimetastatic, poorly differentiated, and anaplastic require other therapies varying from local treatments to systemic therapies. Local treatments are always preferred when the disease is oligometastatic or, if multimetastatic, with only one specific lesion that is growing. When the disease is advanced, multimetastatic, and progressive, the systemic therapy should be initiated and nowadays there are already four different types of TKIs approved; two for MTC and two for RAI refractory DTC and PDTC. The TKIs therapy changed the outcome of advanced metastatic TC patients but, although a prolonged PFS was well documented, an improvement of the OS has yet to be established. Clinicians should bear in mind that TKIs are not curative and a continuative treatment is needed to maintain a response to the disease. For this reason, it is very important that other studies will be performed and other therapeutic strategies will be explored. Moreover, the first generation of TKIs, being multikinase inhibitors, has a lot of side effects that may reduce the quality of life of these patients, frequently leading to a discontinuation of the drug with an important impact on neoplastic disease recurrence. For this reason, new therapeutic approaches are needed and, according to preliminary studies, a new second generation of selective TKIs seems to be a good option able to balance efficacy and AEs in patients with advanced, metastatic, and progressive TC.

## Figures and Tables

**Figure 1 ijms-22-03117-f001:**
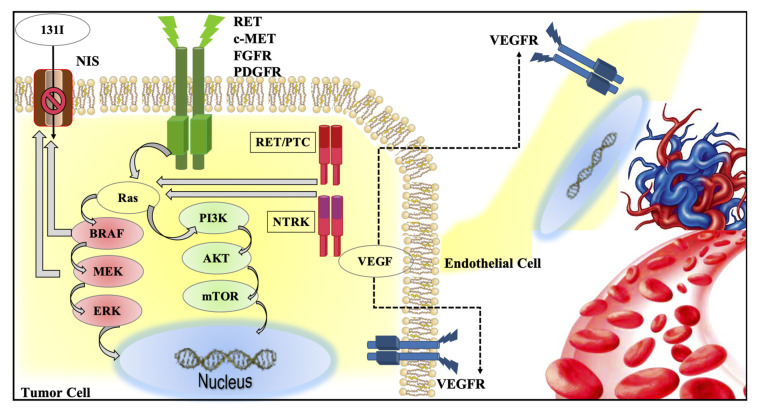
Graphic representation of intracellular pathways (i.e., mitogen-activated protein kinase (MAPK) and phosphoinositide-3-kinase (PI3K)), activated by tyrosine kinase receptors (i.e., RET, MET, FGFR, PDGFR) in thyroid cancer. Activation of *BRAF* and *MEK* also play an important role in the loss of sodium iodide symporter (NIS) activity, leading to RAI refractoriness. The vascular endothelial growth factor receptor is present on the cellular membrane of both tumor and endothelial cells and it is the major player of the new tumor angiogenesis.

**Figure 2 ijms-22-03117-f002:**
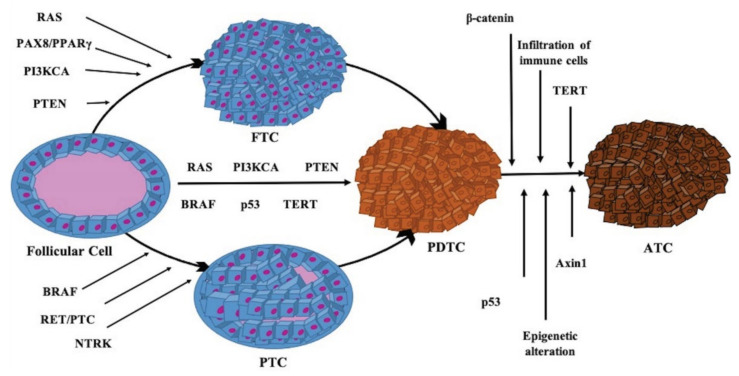
Molecular alterations that lead to mitogen-activated protein kinase (MAPK), phosphoinositide-3-kinase (PI3K), and receptor kinase pathways activation and promote the progression of follicular thyroid cells to papillary (PTC) and to follicular thyroid cancer (FTC). Additional mutations and rearrangements and an increase of MAPK and PI3K pathways signaling promote further progression to poorly differentiated thyroid cancer (PDTC). Further genetic events, especially involving p53, epigenetic alteration, and infiltration of immune cells, promote the onset of anaplastic thyroid cancer (ATC).

**Table 1 ijms-22-03117-t001:** Molecular targets of currently available tyrosine kinase inhibitors.

Drugs/ Clinical Trials	VEGFR-1	VEGFR-2	VEGFR-3	c-KIT	RET	PDGFR	FGFR	EGFR	Others
Lenvatinib (Schlumberger et al.)	+	+	+	+	+	+	+	-	RET-KIF5B rearrangements
Sorafenib (Brose et al.)	-	+	+	+	+	+	-	-	Raf, FLT3
Vandetanib (Wells et al.)	-	+	-	+	+	-	-	+	RET-KIF5B rearrangements
Cabozantinib (Elisei et al.)	-	+	-	+	+	-	-	-	MET, RET-KIF5B rearrangements
Larotrectinib (Drilon et al.)	-	-	-	-	-	-	-	-	TRK1
Entrectinib (Doebele et al.)	-	-	-	-	-	-	-	-	TRK, ALK, ROS1
Selpercatinib (Wirth et al.)	-	-	-	-	+	-	-	-	-
Pralsetinib (NCT03037385)	-	-	-	-	+	-	-	-	-
Vemurafenib (Brose et al.) (Hytman et al.)	-	-	-	-	-	-	-	-	BRAFV600E
Dabrafenib (Falchook et al.)	-	-	-	-	-	-	-	-	BRAFV600E

EGFR: epidermal growth factor receptor; FGFR: fibroblast growth factor receptor; KIT: v-kit Hardy-Zuckerman 4 feline sarcoma viral oncogene; Raf: v-raf murine sarcoma viral oncogene homolog; FLT3: Fms-like tyrosine kinase 3; MET: hepatocyte growth factor receptor; PDGFR: platelet-derived growth factor receptor; RET: REarranged during Transfection receptor; TRK: tropomyosin receptor kinase; VEGFR: vascular endothelial growth factor receptor.

**Table 2 ijms-22-03117-t002:** More frequent adverse events reported for the currently available tyrosine kinase inhibitors.

Adverse Events (All Grade)	Lenvatinib (%)	Sorafenib (%)	Vandetanib (%)	Cabozantinib (%)	Selpercatinib (%)	Pralsetinib (%)	Larotrectinib (%)	Entrectinib (%)
Hypertension	68	41	32	33	43	40	11	NR
Diarrhea	59	69	56	63	38	34	22	35
Skin rash	15	50	45	19	NR	24 *	NR	11
Anorexia	49	32	21	46	NR	15	13	13
Fatigue	59	50	24	41	38	38	37	48
Nausea	41	20	33	43	35	17	29	34
Weight loss	46	47	10	48	NR	NR	NR	NR
QT prolongation	8	NR	14	NR	19	NR	NR	3.1
Hand-foot syndrome	32	76	NR	50	NR	NR	NR	NR
Weight gain	NR	NR	NR	NR	25	NR	15	25
Increased aspartate aminotransferase level	0.4 **	23	NR	86	57	69	45	44
Increased alanine aminotransferase level	0.4 **	26	NR	86	51	43	45	38

Abbreviations: NR: not reported. * Rash includes dermatitis, dermatitis acneiform, eczema, palmar-plantar, erythrodysaesthesia syndrome, rash, rash erythematous, rash macular, rash maculo-papular, rash papular, rash pustular. ** This percentage, the only reported in the drug-related study, is referred only to serious adverse events.

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
