# Peer review of "Thyroid Cancers: From Surgery to Current and Future Systemic Therapies through Their Molecular Identities"

_ijms, 2021, doi:10.3390/ijms22063117_

Round 1

Reviewer 1 Report

Thyroid Cancers: From Surgery to Current and Future Systemic Therapies through Their Molecular Identities manuscript is well-written, comprehensively researched, and logically organised in sections with a detailed description. 

Author Response

We really thank this reviewer for his/her comments. 

Reviewer 2 Report

The manuscript entitled “THYROID CANCERS: FROM SURGERY TO CURRENT AND FUTURE SYSTEMIC 1 THERAPIES THROUGH THEIR MOLECULAR IDENTITIES” by Lorusso et al. is a well written and exhaustive review aiming at taking stock of the situation regarding the available systemic therapies of advanced and progressive thyroid cancers.

I would add more emphasis to predictors that could assist in determining which patients will benefit most from tyrosine kinase inhibitor therapy and to tumor heterogeneity, another confounder when determining the suitability (or resistance) of TKI therapy. In this context it should pointed out the importance that could have the tumor microenvironment in determining the tumor progression and consequently its response to systemic therapies.

Author Response

We thank the reviewer for his/her comments and suggestions. Accordingly with them, we modified the original manuscript adding specific paragraphs about tumor microenvironment and predictive markers/drug resistance. The revised part of the manuscript is highlighted in yellow.

Reviewer 3 Report

The review by Lorusso et al. is focused on the therapies of thyroid cancers. The topic of this review is extremely relevant as testified by the many papers covering these issues published in the last few years (some by the same research group).

Sections of the review are very detailed and well written, but in some cases, in my opinion, some integration could improve the quality of the paper.

Specific comments

  • The authors state “Growing evidences have revealed the importance of tumour microenvironment”. Indeed, the components of the tumor microenvironment play important roles in tumor initiation and progression and could be effectively targeted by immunotherapy. Therefore, a specific paragraph on this relevant topic could be useful.
  • Redifferentiation therapy to increase or restore RAI uptake in tumors should be mentioned.
  • Adverse events are clearly reported in TABLE 2. The authors could omit these data from the text. This will improve the readability  of the paper.

Author Response

Response to Reviewer 3 comments and suggestions:

The review by Lorusso et al. is focused on the therapies of thyroid cancers. The topic of this review is extremely relevant as testified by the many papers covering these issues published in the last few years (some by the same research group).

Sections of the review are very detailed and well written, but in some cases, in my opinion, some integration could improve the quality of the paper.

We really thank this reviewer for his/her positive comments.

Specific comments

Point 1: The authors state “Growing evidences have revealed the importance of tumour microenvironment”. Indeed, the components of the tumor microenvironment play important roles in tumor initiation and progression and could be effectively targeted by immunotherapy. Therefore, a specific paragraph on this relevant topic could be useful.

Thank you for your suggestion. Specific paragraphs on tumor microenvironment and another one on immunotherapy were added at line 276-304 and line 754-793 respectively. The revised part of the manuscript is highlighted in yellow.

Point 2: Redifferentiation therapy to increase or restore RAI uptake in tumors should be mentioned.

Thank you for your comment. A paragraph on re-differentiation therapy was added (lines 623-674). Figure 1 was also modified accordingly. Moreover, to introduce the concept of “loss of differentiation”, we added also a sentence about BRAF mutation (line 220-22). The revised part of the manuscript is highlighted in yellow.

Point 3: Adverse events are clearly reported in TABLE 2. The authors could omit these data from the text. This will improve the readability of the paper.

Thank you for your suggestion. We agree with the reviewer and adverse events were omitted from the text. The revised part of the manuscript is highlighted in yellow.